# Effects of Nitrogen Fertilization and Plant Density on Proso Millet (*Panicum miliaceum* L.) Growth and Yield under Mediterranean Pedoclimatic Conditions

Enrico Palchetti [1,†] , Michele Moretta [1,*,†] , Alessandro Calamai [1], Marco Mancini [1], Matteo Dell'Acqua [2],
Lorenzo Brilli [3] , Paolo Armanasco [1] and Alberto Masoni [1,4]

[1] Department of Agriculture, Food, Environment and Forestry, DAGRI, University of Florence, 50144 Firenze, Italy; enrico.palchetti@unifi.it (E.P.); alessandro.calamai@unifi.it (A.C.); marco.mancini@unifi.it (M.M.); paolo.armanasco@unifi.it (P.A.); alberto.masoni@unifi.it (A.M.)
[2] Institute of Life Sciences, Scuola Superiore Sant'Anna, 56127 Pisa, Italy; matteo.dellacqua@santannapisa.it
[3] Institute for the Bioeconomy, National Research Council of Italy, CNR-IBE, 50145 Firenze, Italy; lorenzo.brilli@ibe.cnr.it
[4] Department of Biology, University of Florence, 50121 Firenze, Italy
[*] Correspondence: michele.moretta@unifi.it
[†] These authors contributed equally to this work.

**Abstract:** In recent years, the dry-land cereal proso millet has become an interesting crop for cultivation in the Mediterranean environment due to the consequences of climate change. It can be considered a resilient crop because it is particularly successful in extreme drought and high-temperature conditions. The goals of this research study were to compare different plant densities (D) and nitrogen fertilization rates (N) in millet (*Panicum miliaceum*), evaluating morphological, productive, and phenological traits. A 2-year field experiment was carried out in Italy, and millet (var. Sunrise) was subjected to four nitrogen fertilization rates (0, 50, 100, and 150 kg N ha$^{-1}$) in interaction with three plant densities (55, 111, and 222 plants m$^2$). Significant differences were found in all the investigated plant traits. The highest grain yield data (i.e., 3.211 kg ha$^{-1}$ and 3.263 kg ha$^{-1}$) and total biomass (i.e., 11.464 kg ha$^{-1}$ and 11.760 kg ha$^{-1}$) were obtained with the N rate of 150 kg ha$^{-1}$ and density of 222 plants m$^2$. Regarding protein content, the highest values were observed using N50, N100, and N150 (ranging from 10.03% to 10.14%) and with D55 (10.43%). Phenological parameters were affected by both plant density and nitrogen amount and decreased when higher levels of these two factors were employed.

**Keywords:** plant density; nitrogen; *Panicum miliaceum*; drought-resistant crop





## 1. Introduction

Proso millet (*Panicum miliaceum* L.) is an annual grain crop mainly cultivated in Asia and Northern Africa for different purposes such as human food, forage, and fuel [1]. In Europe, it is considered a minor cereal due to its low economic relevance, since it is only used there as livestock feed [2]. The term "millets" usually refers to the grains and derived products belonging to a heterogeneous group of highly variable small-seeded grasses (e.g., *Cenchrus americanus*, *Setaria italica*, *Pennisetum glaucum*, and *Paspalum scrobiculatum*) with different origins, taxonomies, and cultivation distributions. Proso millet is recognized as one of the oldest domesticated cereal crops in the world, dating to more than 10,000 years ago, and it represents one of the most important crops for smallholder farmers' subsistence in the semiarid lands of several tropic countries [2–4]. Despite the limited yield potential in conventional agriculture (1.4 t ha$^{-1}$ and 3.6 t ha$^{-1}$), the high crop growth rate, heat stress tolerance, large leaf area index, and high water efficiency confer good adaptability to proso millet plants under different environmental stresses exacerbated by climate change [5,6]. In addition, it has a shorter growing season (60–90 days) compared to all major cereals, and

this actively contributes to its drought tolerance because it can be sown and harvested while avoiding extreme weather conditions. Furthermore, it can be described as an excellent intercropping crop, good for crop rotation in Mediterranean environments. In this area, under a cropping system with summer fallow and winter wheat rotation, proso millet could replace this fallow if planted in early spring, helping to maintain the soil organic matter [7]. Millet covers the soil surface, minimizing the biological oxidation of the topsoil layer and helping to maintain the deep soil moisture to be used by the cultivated crops that follow. At the same time, thanks to its C4 metabolism, low transpiration ratio, and shallow root system, it can efficiently fix carbon under the stressful conditions that characterize the summer period [8]. Synergism with proso millet facilitates weed control of both annual and perennial broadleaf weeds; their density declines by nearly 90% if millet is followed by winter crop cultivation for two consecutive agronomic seasons [9].

These favorable traits may play an important role in promoting sustainable agricultural food systems, facilitating agroecosystems crop diversity, and making this species a climate-smart crop [10–13]. Millet-based foods also have a beneficial effect on human health [4] thanks to their reduced glycemic index, high amount of fiber, and absence of gluten protein, which makes them suitable also for people with celiac disease [8,14]. Furthermore, compared to the major cereals, millet grains have a higher content of essential amino acids (e.g., cysteine and methionine a), vitamins, protein content (i.e., 12.5%), dietary fibers, resistant starches, and micronutrients [14,15]. However, despite its health benefits and valuable nutritional composition, worldwide cultivation of proso millet in the last decade has declined, probably due to its low grain yield (world mean yield of nearly 1 ton ha$^{-1}$) [16], the scant knowledge of its cultivation [17,18], and the scarce varieties available due to it being overlooked by modern breeding programs [19–21].

Despite more than 150,000 accessions stored in gene banks worldwide, millet breeding improvement has been carried out only in a few parts of the world (China, India, the USA, and Russia), with limited success [22]. Genetic diversity assessment in proso millet is challenging; it has been explored only to a limited extent, and only seldom have next generation sequencing (NGS) technologies been used [20]. The lack of sufficient genetic characterization of the available germplasm and the natural cross-pollination (more than 10%) makes the breeding program a very challenging activity [8]. However the recently published proso genome sequence and chromosome assembly in 2019 projected this species into the modern genomics era, disclosing a great potential for large-scale genotyping applications and, hence, future molecular-assisted breeding programs [23]. A few studies indicated that *P. miliaceum* crop performance was particularly affected by two management practices, nitrogen fertilization (N) and plant density (D), adopted during the growing cycle [24,25]. Nitrogen seems to be the major element required to boost the potential yield; studies revealed that the optimum range may vary between 50 and 150 N kg ha$^{-1}$, according to pedoclimatic conditions [17,26]. However, no studies on this topic have been carried out in Europe. At the same time, optimum plant density promotes better growth and the best conditions for plants, exploiting moisture, nutrients, and solar radiation [7].

In this study, we evaluated the effects of different levels of nitrogen fertilization (N) and plant density (D) on millet plants' morphological, productive, and phenological traits under a Mediterranean pedoclimatic environment. Three plant densities and three levels of fertilization were tested over two years of field experiments to evaluate the best combination to improve millet agronomic performances. Considering the future water scarcity scenario that will afflict all Mediterranean areas, it is crucial to employ resilient crops and to clarify the possible role of agronomic practices to mitigate this phenomenon and provide food security.

## 2. Materials and Methods

### 2.1. Study Area

Millet cultivation was studied for two consecutive years (2018 and 2019) in the study area located on a farm belonging to the Tuscan Regional Administration located in Cesa

(Central Italy, 43°18′32″ N; 11°49′35″ E). The climate is typically Mediterranean, with a long-term yearly precipitation average of about 700 mm mainly distributed in spring and autumn, and monthly minimum and maximum temperatures recorded in January (6 °C) and July (25 °C), respectively. For the two years of the experiment, weather data were collected by a weather station located near the field and reported on a monthly scale in Figure S1. The experimental soil characteristics are listed in the table below (Table 1).

**Table 1.** Chemical and physical traits of the experimental soil.

| Properties | Units | Value |
|---|---|---|
| Sand | % | 25.4 |
| Silt | % | 30.1 |
| Clay | % | 44.5 |
| Total organic matter | % | 1.66 |
| Total nitrogen | % | 0.12 |
| Available phosphorous | ppm | 11 |
| pH | | 7.1 |
| Electrical conductivity (EC) | mS cm$^{-1}$ | 0.154 |
| Cation exchange capacity (CEC) | meq 100 g$^{-1}$ | 27.46 |
| Ca exchangeable | ppm | 21.25 |
| Mg exchangeable | ppm | 5.17 |
| Na exchangeable | ppm | 0.58 |
| K exchangeable | ppm | 0.46 |

*2.2. Experimental Design*

The experimental design was a strip-plot design replicated two times (two blocks) for two consecutive years. Each experimental plot (n = 24) was 7.2 m$^2$ (2.4 m × 3 m), and a combination of four N levels (0, 50, 100, and 150 kg$^{-1}$) and three planting densities (i.e., 55 plants m$^2$, 111 plants m$^2$, and 222 plants m$^2$) were tested. In both years, the precedent crop cultivated in the experimental fields was bread wheat (*Triticum aestivum*). After winter tillage at a depth of 25 cm using a disc harrow, the seedbed was prepared with a spike-tooth harrowing at 6–8 cm depth just before seeding. All fertilization levels were applied as a single dose at sowing time using nitrogen (urea, N > 46%). The proso millet variety Sunrise was sown on 4 May and harvested on 25th August in 2018, and on 9 May and 30 August in the 2019 season. We chose this variety because it was the most common variety commercialized in Italy. Plots were manually weeded as required during crop growth, without the use of pesticides. Since the purpose of our experiment was to evaluate the feasibility and the effect of D and N factors in millet cultivation under natural semiarid conditions, no irrigation was administered. Finally, no parasitic attacks occurred in either year that would require phytosanitary treatments; therefore, no treatments were carried out.

*2.3. Data Collection and Statistical Analysis*

At harvest time, defined according to the BBCH (Biologische Bundesanstalt, Bundessortenamt, und Chemische Industrie) phenological scale [27], three plants were randomly collected from each plot and were measured to evaluate the following morphological traits: plant height (dis), basal tiller, leaf number, seed yield per plant, inflorescence length, and peduncle length. The productive traits, such as grain yield, total plant biomass, and 1000-seed weight, were evaluated instead, for each plot, by harvesting a standard harvest square of 0.50 m$^2$ area and oven-drying at 55 °C for 48 h to reduce the moisture content. Furthermore, the harvest index (HI, grain yield/total biomass) [18] and days to flowering (time from the emergence until at least 50% flowering plants in a plot), and days to maturity (time from the emergence to the harvest cultivation stage according to the cereal phenological BBCH scale) were calculated, and the latter two parameters were expressed as days after emergence (DAE). Protein grain content was determined using standard international Kjendahl Method, AACC 46-10 (conversion factor N × 5.83) [28].

Analysis of variance was carried out by fitting a generalized linear mixed model (GLMM), considering the block as a random factor and the year and assigned agronomic treatments (D and N) as fixed factors, both alone and in their interactions. A post hoc Tukey test was carried out for multiple comparison tests among levels of each factor. For each measured trait, only the significant source of variation was discussed. Statistical analysis was performed using R software v.4.2.2 (R Core Team, 2019) with the R/packages "lme4" and "multcomp". In addition, linear regression models were fitted to evaluate the effect of N rates on plant morphology (plant height), production (seed yield per plant and plant biomass), and phenology (days to flowering) for each year using the function "lm()". For all regressions, goodness of fit was given as R-squared values ($R^2$) [29].

## 3. Results

The results of the statistical analysis (Table 2) indicated that the year factor had significant effects on nearly all the traits analyzed, except for 1000-seed weight, HI, and days to maturity ($p > 0.05$). Significant differences in morphological parameters were observed in the year 2019 compared to the first year of the trial. The plant density treatments adopted during the field tests significantly influenced all the parameters evaluated in 2018 for plant height (92.03 cm and 89.35 cm, respectively), basal tillers (5.60 and 5.20, respectively), seed yield per plant (220.86 g and 206.87 g, respectively), peduncle length (107.71 mm and 105.85 mm, respectively), and inflorescence length (221.67 mm and 215.75 mm, respectively) (Table 2). The same trend was also found for productive parameters, where proso millet cultivated during the year 2019 showed greater grain yield, total biomass, and protein content (3147 kg ha$^{-1}$, 11,526 kg ha$^{-1}$, and 10.26%, respectively) compared to the year 2018 (2847 kg ha$^{-1}$, 10,566 kg ha$^{-1}$, and 9.97%, respectively). Regarding phenological parameters, millet took more time (i.e., 7.5 days, on average) to flower in 2018 than in 2019, although plots cultivated in both years did not show differences in days to maturity, reaching physiological maturity at 99 DAE.

**Table 2.** Analysis of variance for productive, morphological, and phenological parameters of millet. Trait means followed by the same letters among the factor levels are not significant at $p > 0.05$, while values followed by different letters are significant at $p < 0.05$. The "*" represents the different significance levels of each factor in the model; ns = $p > 0.05$, "*" = $p < 0.05$, "**" = $p < 0.01$.

| | Morphological Data | | | | | Productive Data | | | | | Phenological Data | |
| | Plant Height (cm) | Basal Tiller (n) | Peduncle Length (mm) | Inflorescence Length (mm) | Seed Yield per Plant (g) | 1000 Seed Weight (g) | Grain Yield (kg) | Total Biomass (kg) | Harvest Index | Protein (%) | Days to Flowering | Days to Maturity |
|---|---|---|---|---|---|---|---|---|---|---|---|---|
| Year (Y) | ** | * | ** | ** | ** | ns | ** | ** | ns | * | ** | ns |
| 2018 | 89.35 b | 5.20 b | 105.85 b | 215.78 b | 206.87 b | 6.46 | 2847 b | 10,565 b | 0.26 | 9.77 b | 62.70 a | 99.1 |
| 2019 | 92.03 a | 5.60 a | 107.71 a | 221.67 a | 220.86 a | 6.85 | 3147 a | 11,525 a | 0.27 | 10.26 a | 55.20 b | 99.4 |
| Plant density (D) | * | * | * | ** | ** | ns | ** | ** | ** | ** | ** | ** |
| D55 | 91.73 a | 6.45 a | 110.04 a | 230.29 a | 236.55 a | 6.77 | 2758 c | 10,559 b | 0.26 c | 10.43 a | 61.10 a | 102.30 a |
| D111 | 90.19 b | 4.91 b | 107.31 b | 215.21 b | 206.73 b | 6.67 | 3020 b | 11,113 ab | 0.27 b | 9.81 b | 57.90 b | 98.80 b |
| D222 | 90.13 b | 4.85 b | 102.98 c | 210.67 c | 198.32 c | 6.49 | 3211 a | 11,464 a | 0.28 a | 9.79 b | 56.30 c | 96.20 c |
| Nitrogen (N) | ** | ** | ** | ** | ** | ns | ** | ** | ** | ** | ** | ** |
| N0 | 87.06 c | 3.86 c | 97.39 b | 207.89 c | 193.32 c | 6.46 | 2705 d | 10,668 c | 0.25 c | 9.78 b | 60.50 a | 101.20 a |
| N50 | 88.32 c | 3.75 c | 101.36 b | 215.06 b | 205.33 b | 6.62 | 2896 c | 10,655 c | 0.27 b | 10.03 a | 59.30 a | 98.50 b |
| N100 | 92.67 ab | 6.67 b | 112.50 a | 224.28 a | 227.73 a | 6.71 | 3123 b | 11,101 b | 0.27 b | 10.11 a | 57.10 b | 97.20 b |
| N150 | 94.72 a | 7.34 a | 115.86 a | 227.67 a | 229.08 a | 6.78 | 3263 a | 11,760 a | 0.28 a | 10.14 a | 55.90 b | 95.70 c |
| D × N | ** | ns | ** | ** | ** | ns | ** | ** | ** | ** | ** | ** |
| D × Y | ns | ns | ns | ns | * | ns | * | ** | * | ns | ns | ns |
| N × Y | ns | ns | ns | ns | ns | ns | * | * | ** | ns | ns | ns |

The plant density treatments adopted during the field trial significantly influenced all the parameters evaluated ($p < 0.05$), except for 1000-seed weight ($p > 0.05$) (Table 2). Increasing plant density from 55 to 222 plants m$^2$ induced a progressive decrease in several traits analyzed (Table 2), including plant height, number of basal tillers, seed yield per plant, peduncle length, inflorescence length, and protein content. Differences caused by plant density were also observed in *P. miliaceum* phenology. Plots cultivated at low density took slightly more time to flower and to reach physiological maturity (D 55; 61

and 102 DAE, respectively) compared with plots maintained at intermediate (D111; 58 and 99 DAE) and high densities (D222; 56 and 96 DAE). However, an opposite trend was scored considering productive traits; grain yield, total biomass, and HI progressively and consistently increased with plant density from D55 to D111 (+9.5%, +5.3%, and +3.8%, respectively) and from D55 to D222 (+16.5%, +8.6%, and 7.7%, respectively).

The field experiment showed a clear effect of nitrogen fertilization rates on close morphological, productive, and phenological data. The application of N significantly affected protein content ($p < 0.05$) and all the other parameters ($p < 0.01$), with the only exception being 1000-seed weight ($p > 0.05$) (Table 2). The shortest plants with a low number of basal tillers were recorded at the control level (N0; 87.1 cm and 3.9, respectively) and at N50 (88.3 cm and 3.7, respectively), while the N150 level showed the maximum plant height (94.7 cm) and the higher basal tiller number (7.3). At the same time, seed yield per plant and inflorescence length showed the same increasing pattern: 193.3 g and 207.9 mm at control (i.e., N0); 205.3 g and 215.1 mm using N50; and 115.9 g and 229.1 mm when N150 was distributed. Plants' peduncle length was the same over N0 and N50 application (97.4 and 101.4 mm, respectively), as well as over N100 and N150, which showed similar but higher data values (112.5 and 115.9 mm, respectively). Regarding productive data, nitrogen application had positive effects on all proso millet parameters. For example, N0 gave the lowest grain yield (2705 kg ha$^{-1}$), total biomass (10,668 kg ha$^{-1}$), and HI (0.25), while the N50 application reported an increase only in grain yield (+7%) and HI (+8%). At the same time, compared to N50, N150 gave the highest grain yield (+20.6%), total biomass (+10.2%), and HI (+12%), followed by N100 (15.5%, 4%, and +8%, respectively).

Interaction between plant density and nitrogen fertilization significantly affected all the parameters analyzed except the number of basal tillers and 1000-seed weight (Table 2). Interaction between years with both agronomic factors (D, N) did not seem to affect morphological and phenological traits; instead, they significantly influenced all the productive parameters except protein content (Table 2).

Results of linear regression analyses underlined a positive relation between N fertilization and plant morphology (Figure 1a) and productive traits (Figure 1b,c). An increase in N amount from the lower level (50) to the higher (100–150) corresponds to an increase in plant height, seed yield, and total biomass, but the increased amount is not proportional to the N doses applied. Regarding the plant phenology (Figure 1d), days to flowering were negatively related to the N rates, and the flowering period was anticipated till up to 3–4 days with the increase in N amount applied. All the linear regression curves fitted have high values of R$^2$ (Figure 1), ranging from 0.58 to 0.93, underlining the goodness of the different fitting processes.

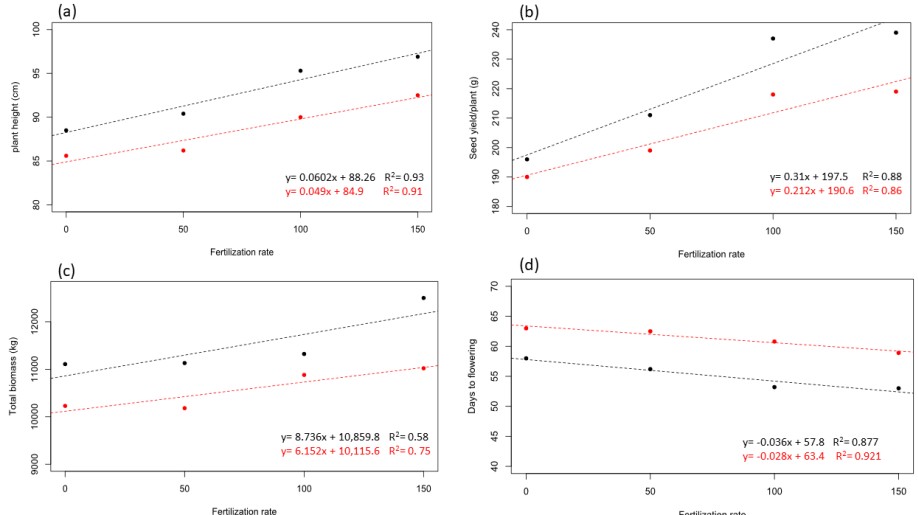

**Figure 1.** Linear regression analysis chart between N fertilization rates and plant height (**a**), seed yield per plant (**b**), total biomass (**c**), and days to flowering (**d**). Dot indicates mean values of the traits

recorded for each fertilization level, while dot colors indicate the agronomic season: red for 2018 and black for 2019. Each graph also reports the interpolated fitted dotted lines, the lines' equation, and $R^2$ values (red and black colors according to the 2018 and 2019 data, respectively).

## 4. Discussion

Due to low productivity, the cultivation of red millet in Western Europe is limited, and there is a lack of information on its agronomic management. However, its nutraceutical properties, stress tolerance, and high water efficiency have prompted renewed interest in its cultivation in crop rotation systems, given future climate change. Aiming to help fill this gap, our study evaluated the agronomic performance of *P. miliacum* var. Sunrise under different cultivation practices in Mediterranean soil and climate conditions in two consecutive growing seasons (2018 and 2019). Different plant densities (D) and nitrogen fertilization rates (N) were applied to evaluate their effects on several agronomic traits (i.e., morphological, productive, and phenological) and plant performances. The identification of the best site-specific combination of D and N represents important information to improve crop grain yield and counteract the effects of climate change, reducing GHG emissions and nutrient leaching into the groundwater [30]. Overall, the present results showed that the grain yield of proso millet was slightly lower to that previously obtained in the Mediterranean area over a 3-year study conducted in Turkey (3200 kg ha$^{-1}$) and a 2-year experiment conducted in Italy (3125 kg ha$^{-1}$), but higher than those recorded in the United States (2016 kg ha$^{-1}$) and India (2600 kg ha$^{-1}$) [17,31,32]. Gong et al. [24] surprisingly obtained grain yields ranging between 3500 and 4600 kg ha$^{-1}$, cultivating proso millet variety Youmi 2 in a semiarid area in China characterized by continental monsoon climate with an annual mean precipitation of 400 mm and mean temperature of 8.3 °C. However, data confirmed the lower grain yields of *P. miliaceum* compared with the major spring–summer cereal crops grown in the Mediterranean region, which varied from 4710 kg ha$^{-1}$ to 5910 kg ha$^{-1}$ for sorghum (*Sorghum bicolor* (L.) Moench) and from 7110 kg ha$^{-1}$ to 8970 kg ha$^{-1}$ for maize (*Zea mays* L.), confirming that extensive research is needed on the development of breeding lines for crop improvement [33].

Field experiment results showed better productive traits in the year 2019 compared with those recorded in 2018. The yield reduction recorded in the year 2018 was probably related to the unusually high precipitation rate recorded in July 2019 and to a short drought period occurring during the ear-emergence stage, which negatively affected some morphological traits that drive *P. miliaceum* growth [11,34]. In particular, the low yield seems to be the result of a reduction in photosynthesis efficiency and a decrease in the seed yield per plant due to the flower's abortion [35]. Similar results have been previously observed on other cereal crops such as pearl millet (*Pennisetum glaucum*) under comparable cultivation stressful conditions [36,37]. Differences among years were also observed in plant phenology, with the lengthening in the flowering stage of 1 week during the year 2018 compared with that of the following year. Earlier studies recording different days to flowering in a multiyear experimental field of millet suggested that changes in the length of some phenological phases may be a mechanism of drought tolerance [38]. This strategy to cope with drought by adjusting phenology duration according to rainfall pattern is well-known for sorghum (*Sorghum bicolor* (L.) Moench). This ability is considered an imperative to accelerate progress in the development of climate-resistant crops, which could ensure production even under extreme climatic conditions [39,40].

The protein content measured in both experimental years was within the range 5.6% to 15.62% reported in the literature, depending on the variety [19,41]. Generally, in cereal crops, high grain yields were inversely related to grain protein concentrations due to the dilution of nitrogen when carbohydrate deposition increases [42]. In contrast, in the present study, the occurrence of a higher rainfall during the most productive year (2019) may have

extended the grain-filling duration and favored the nitrogen uptake to the grain, promoting a greater seed protein accumulation [43].

In accordance with other studies on millet species, our results confirmed the nitro-positive effect of fertilization on improving biomass allocation, which leads to an improvement in morphological traits in *P. miliaceum* [24,26,41]. Although the yield and quality of cereal grain are mainly influenced by N soil availability and N fertilizer distribution, limited soil water availability could reduce the N-positive effects on crops [17,33]. As a consequence, plants cultivated in different pedoclimatic conditions respond differently to N fertilization [44]. For example, authors have conducted site-specific studies on millets in Pakistan and Turkey, recommending 60 and 120 kg N ha$^{-1}$ dosages of nitrogen application, respectively [17,45]. In the present study, results showed that N application at an increasing rate of up to 150 kg N ha$^{-1}$ produced taller plants with high basal tillers, which are appreciated as they contributed to intercepting more solar radiation, improving leaf area index (LAI), and, in turn, increasing grain yield and total biomass [46]. Despite higher doses of N-increased morphological and productive traits, authors have noted that excessive N fertilization could promote the tilting and bending of plant stems [7,47]. Plant lodging in cereal represents a known problem during the harvest process and could result in a potential health risk for humans and animals due to mycotoxin accumulation in grains [48]. However, no lodging effects were observed during our experiment.

Regarding the density rate, increasing the number of plants may be an effective strategy for improving proso millet productivity, according to the levels we tested. Optimum plant density, indeed, promotes better growth and the best conditions for plants to exploit moisture, nutrients, and solar radiation [7]. Total grain yield increases accordingly with the density rate, and this is not a foregone behavior under semiarid conditions [49,50]. High density levels usually correspond to higher plant competition for key input resources such as radiation, water, and nutrients, and this negatively affects millet plant morphology growth [51]. A higher plant density reduced plant photosynthetic capacity, stomatal conductance, and leaf chlorophyll content but at the same time increased mulching of the soil surface, improved crop evapotranspiration, reduced radiation from the soil surface, and reduced soil water evaporation [52]. Therefore, a higher number of plants often causes the decline in crop morphology and productivity of every single plant but improves total yield and favors its stability due to a greater number of main stems per unit area [53,54], and this seems to be our case. Higher amounts of plants significantly influenced also the phenological traits of proso millet, causing a reduction in the flowering and maturity time according to the higher density rate. In our case, it seems that a reduction in plant morphology for plant competition may correspond to a shorter growing cycle; however, the effects of climate and water availability cannot be excluded [34]. Finally, the identification of the optimal plant density for high yield production relied also on the cultivars chosen and the cultivation environments [55].

## 5. Conclusions

In recent years, proso millet has stimulated great interest for its possible role as a renewal crop contributing to crop diversification in sustainable rotational crop programs. Its short growing season is a great advantage for farmers in the Tuscan semiarid zones, who may cultivate it during spring and then harvest postsummer, preparing the soil for winter crop cultivation.

Two-year results obtained from this research have revealed that *Panicum miliaceum* can be cultivated in Italy with a good yield and grain quality. An amount of 150 kg N ha$^{-1}$ nitrogen fertilization and 222 plant m$^{-2}$ were the combinations recommended in soils poor in organic matter to obtain the best cultivation results.

**Supplementary Materials:** The following supporting information can be downloaded at: https://www.mdpi.com/article/10.3390/agriculture13091657/s1, Figure S1: Monthly air temperatures (maximum and minimum) and precipitation for the years 2018 (a) and 2019 (b) recorded in the study area.

**Author Contributions:** Conceptualization, M.D.; data curation, M.M. (Michele Moretta) and M.D.; formal analysis, M.M. (Michele Moretta) and A.M.; investigation, A.C. and M.M. (Marco Mancini); methodology, M.M. (Michele Moretta) and A.M.; project administration, E.P.; supervision, E.P. and A.M.; validation, A.M.; visualization, A.C., M.M. (Marco Mancini), P.A. and L.B.; writing—original draft, E.P., M.M. (Michele Moretta), P.A. and L.B. All authors have read and agreed to the published version of the manuscript.

**Funding:** This research received no external funding.

**Institutional Review Board Statement:** Not applicable.

**Informed Consent Statement:** Not applicable.

**Data Availability Statement:** Data available on request from the authors.

**Conflicts of Interest:** The authors declare no conflict of interest.

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
