# Peer review of "Effects of Nitrogen Fertilization and Plant Density on Proso Millet (Panicum miliaceum L.) Growth and Yield under Mediterranean Pedoclimatic Conditions"

_agriculture, doi:10.3390/agriculture13091657_

Round 1
Reviewer 1 Report
In this research "Effects of nitrogen fertilization and plant density on proso millet (Panicum miliaceum L.) growth and yield under Mediterranean pedoclimatic conditions", the subject of this experiment is interesting, however, there are many minor and major issues. Overall, I hope that the authors will solve these issues in the manuscript, improve the quality and make it suitable for publication in Agriculture.
· The manuscript has minor grammatical faults. Proofread the MS for English improvement.
· In the Introduction section, the authors have the least explanation about the topic of this research. It is recommended to rewrite this section.
- In the Materials and Methods section, please explain:
- What type of nitrogen fertilizer (urea, ammonium nitrate, etc.) was used?
- Determine the time of fertilizing the farm.
- How many times was fertilization done?
· Why was accumulation of nitrates not evaluated?
· Please check and edit references number 8, 38 and 39.
The manuscript has minor grammatical faults. Proofread the MS for English improvement.
Reviewer 2 Report
Comments:
In the work “Effects of nitrogen fertilization and plant density on proso millet (Panicum miliaceum L.) growth and yield under Mediterranean pedoclimatic conditions” (agriculture-2528829) by Enrico et al, the authors have tried to evaluate agronomical management, i.e., four nitrogen fertilization rates, and three planting density on the productivity of proso millet fields. The study is interesting and falls within the journal's scope well. The manuscript can be improved in some phrases, e.g., the rainfall during the growing season, an annual precipitation is not enough for the readers to know the drought conditions.
Major issues:
i. Because the study area is a Mediterranean type, which means 9 May to 30 August (2019) may be a water-limited period. If soil water cannot be satisfying the plant’s growth, some interaction effects may happen with nitrogen and density, or year.
ii. There are three plant density and four nitrogen fertilization, can you show a field photo? Three plant density may give the readers a chance to know the experimental design and, the high nitrogen fertilization the much greener of the crop. This photo should be listed as Figure 1.
Small points:
Line 40: 10,000 years ago, there should be a comma?
Line 71: change “find” into “evaluate”.
Line 80: if the rainfall is mainly distributed in spring and autumn, there probably occurs a drought during the experiments.
Line 154-155: “The field experiment showed a close positive relationship between morphological, productive, phenological data and nitrogen fertilization rates”, can you show the data? You may need to pick some representative traits to regress them to establish “a close positive relationship”, a new figure can be shown in the text.
Line 232: indeed, drought is an un-quantitive factor in this study.
Comments:
In the work “Effects of nitrogen fertilization and plant density on proso millet (Panicum miliaceum L.) growth and yield under Mediterranean pedoclimatic conditions” (agriculture-2528829) by Enrico et al, the authors have tried to evaluate agronomical management, i.e., four nitrogen fertilization rates, and three planting density on the productivity of proso millet fields. The study is interesting and falls within the journal's scope well. The manuscript can be improved in some phrases, e.g., the rainfall during the growing season, an annual precipitation is not enough for the readers to know the drought conditions.
Major issues:
i. Because the study area is a Mediterranean type, which means 9 May to 30 August (2019) may be a water-limited period. If soil water cannot be satisfying the plant’s growth, some interaction effects may happen with nitrogen and density, or year.
ii. There are three plant density and four nitrogen fertilization, can you show a field photo? Three plant density may give the readers a chance to know the experimental design and, the high nitrogen fertilization the much greener of the crop. This photo should be listed as Figure 1.
Small points:
Line 40: 10,000 years ago, there should be a comma?
Line 71: change “find” into “evaluate”.
Line 80: if the rainfall is mainly distributed in spring and autumn, there probably occurs a drought during the experiments.
Line 154-155: “The field experiment showed a close positive relationship between morphological, productive, phenological data and nitrogen fertilization rates”, can you show the data? You may need to pick some representative traits to regress them to establish “a close positive relationship”, a new figure can be shown in the text.
Line 232: indeed, drought is an un-quantitive factor in this study.
Round 2
Reviewer 1 Report
I would like to thank the authors for addressing the suggested comments. It is now ready to be published in Agriculture journal.